

# Foliar mycoendophytome of an endemic plant of the Mediterranean biome (*Myrtus communis*) reveals the dominance of basidiomycete woody saprotrophs

Aline Bruna M. Vaz[1], Paula Luize C. Fonseca[1], Felipe F. Silva[2],
Gabriel Quintanilha-Peixoto[2], Inmaculada Sampedro[3], Jose A. Siles[3],
Anderson Carmo[4], Rodrigo B. Kato[2], Vasco Azevedo[4],
Fernanda Badotti[5], Juan A. Ocampo[3], Carlos A. Rosa[1] and
Aristóteles Góes-Neto[1]

[1] Department of Microbiology, Universidade Federal de Minas Gerais, Belo Horizonte, Minas
Gerais, Brazil
[2] Graduate Program of Bioinformatics, Universidade Federal de Minas Gerais, Belo Horizonte,
Minas Gerais, Brazil
[3] Department of Soil Microbiology and Symbiotic Systems, Estación Experimental del Zaidín,
C.S.I.C., Granada, Spain
[4] Department of Genetics, Ecology, and Evolution, Universidade Federal de Minas Gerais,
Belo Horizonte, Minas Gerais, Brazil
[5] Department of Chemistry, Centro Federal de Educação Tecnológica de Minas Gerais,
Belo Horizonte, Minas Gerais, Brazil

Corresponding author
Aristóteles Góes-Neto,
arigoesneto@icb.ufmg.br

## ABSTRACT

The true myrtle, *Myrtus communis*, is a small perennial evergreen tree that occurs in Europe, Africa, and Asia with a circum-Mediterranean geographic distribution. Unfortunately, the Mediterranean Forests, where *M. communis* occurs, are critically endangered and are currently restricted to small fragmented areas in protected conservation units. In the present work, we performed, for the first time, a metabarcoding study on the spatial variation of fungal community structure in the foliar endophytome of this endemic plant of the Mediterranean biome, using bipartite network analysis as a model. The local bipartite network of *Myrtus communis* individuals and their foliar endophytic fungi is very low connected, with low nestedness, and moderately high specialization and modularity. Similar network patterns were also retrieved in both culture-dependent and amplicon metagenomics of foliar endophytes in distinct arboreal hosts in varied biomes. Furthermore, the majority of putative fungal endophytes species were basidiomycete woody saprotrophs of the orders Polyporales, Agaricales, and Hymenochaetales. Altogether, these findings suggest a possible adaptation of these wood-decaying fungi to cope with moisture limitation and spatial scarcity of their primary substrate (dead wood), which are totally consistent with the predictions of the viaphytism hypothesis that wood-decomposing fungi inhabit the internal leaf tissue of forest trees in order to enhance dispersal to substrates on the forest floor, by using leaves as vectors and as refugia, during periods of environmental stress.

## INTRODUCTION

Mediterranean Forests, Woodlands and Scrubs comprise a distinct biome of the Palearctic Biogeographic Realm (*Olson et al., 2001*). Amongst the distinct ecoregions of this biome, the Southwestern Mediterranean sclerophyllous and mixed forests ecoregion is characterized by old crystalline substrates, such as granite, quartzite, and marble, with hot and dry summers and relatively mild and humid winters (*Olson et al., 2001*). These forests are mainly composed of evergreen broadleaves trees and shrubs such oaks, true myrtle (*Myrtus communis*), laurel and even two endemic palms, which give a unique subtropical feature to these dry, warm coastal landscapes occurring all around the Mediterranean Sea (*Regato, 2001*). Unfortunately, nowadays, these forests, as well as others in the Mediterranean biome, are critically endangered and only remain in small, fragmented areas, such as protected conservation units in the distinct Mediterranean countries (*Regato, 2001*).

*Myrtus communis*, the true (or common) myrtle, is an evergreen perennial and sclerophyll shrub or small tree, usually 1.8–3.0 m in height, with dark green glossy, glabrous and coriaceous leaves, white flowers and blue-black berry fruits (*Sumbul et al., 2011*). It is a diploid plant, which is allogamous and self-compatible, whose fruits are mainly dispersed by birds and small mammals (*Migliore et al., 2012*). *M. communis* is the only species of the tribe Myrtae of the family Myrtaceae that occurs in Europe (*Vasconcelos et al., 2017*), with a typical circum-Mediterranean geographic distribution (*Migliore et al., 2012*).

Besides its extensive use in ethnomedicine for the treatment of disorders such as diarrhea, peptic ulcer, hemorrhoids, inflammation, pulmonary and skin diseases (*Alipour, Dashti & Hosseinzadeh, 2014*), myrtle has also been used in food (liquors, meat and sauces flavor) and cosmetic (perfumes) industries (*Aleksic & Knezevic, 2014*). *M. communis* leaves are rich in terpenes, phenolic acids, tannins, and flavonoids, and its extracts exhibit high antibacterial activity (*Aleksic & Knezevic, 2014*). Furthermore, myrtucommulones (a group of oligomeric nonprenylated acylphloroglucinols) reported from *M. communis* with potent antioxidant, anti-inflammatory, and antineoplastic properties, are also produced by a strain of fungal endophyte isolated from the phyllosphere of this endemic plant of the Mediterranean biome (*Nicoletti et al., 2014*).

The phyllosphere of land plants supports a great richness and abundance of microorganisms and, amongst them, fungal epiphytes and endophytes (*Vorholt, 2012*). Fungal endophytes are internal colonizers of aboveground tissues of plants and, apparently, do not cause any symptoms of diseases on their hosts (*Skaltsas et al., 2019*; *Vaz et al., 2018*). Moreover, visually healthy leaves contain numerous, independent infections, rather than systemic or otherwise extensive growth of hyphae (*Arnold, 2008*).

Fungal endophytes were first reported by the botanist Heinrich Friedrich Link still in the beginning of the 19th century (*Hardoim et al., 2015*); however, only since the 1970's (*Carroll & Carroll, 1978*), 1980's (*Carroll, 1988*) and beginning of the 1990's (*Petrini, 1991*) that they have been more intensively studied. In a very recent and comprehensive review, *Harrison & Griffin (2020)* estimated that, among the 17 biomes

of the Earth (*Olson et al., 2001*), seven were understudied, and together composed only 7% of the studies that were evaluated. One of these understudied biomes is the Mediterranean Forests, Woodlands and Scrubs, where the species *Myrtus communis* typically occurs. This aforementioned systematic review study also pointed out that that fungal endophyte diversity has already been characterized in, at least, one host from 30% of embryophyte families (*Harrison & Griffin, 2020*), including many species of Myrtaceae in the tribe Myrtae, such as *Myrciaria floribunda*, *Eugenia* aff. *bimarginata* (*Vaz et al., 2012*), *Luma apiculata*, and *Myrceugenia ovata* var. *nanophylla* (*Vaz et al., 2014a*, *2014b*).

Fungal endophytes are an omnipresent and phylogenetically diverse group of organisms that establish stable long-term interactions with their plant hosts (*Rashmi, Kushveer & Sarma, 2019*). Furthermore, their impacts (either positive or negative) on the plants where they live at least part of their life cycle, may vary depending on the physiological status of the host, nutrient availability, environmental conditions and interaction with the microbiome and the plant host itself (*Fesel & Zuccaro, 2016*). The impact of fungal endophytes is usually considered as strongly context-dependent (*Rodriguez et al., 2009*), and the relationships between plant hosts and their fungal endophytes can range from mutualism through commensalism to latent or even mild antagonism (*Saikkonen et al., 1998*; *Schulz & Boyle, 2005*; *Porras-Alfaro & Bayman, 2011*). Therefore, many fungal endophytes can, in fact, be latent pathogens and latent saprotrophs (*Hyde & Soytong, 2008*).

Until quite recently, most of the studies on fungal endophytes are still based on a culture-dependent approach (*Christian, Whitaker & Clay, 2017*). Nevertheless, methods based on culture are very selective and highly influenced by the composition of the culture media, the physiological adaptations of the fungi, and the sampling procedures, which influence the taxonomic composition, richness, and abundance of fungal endophytes recovered (*Stone, Polishook & White, 2004*). Therefore, a metagenomics approach, based on the amplification of a taxonomic biomarker before massively parallel sequencing, theoretically provides a significantly more detailed access to the diversity of the mycobiome of any kind of substrate (*Cuadros-Orellana et al., 2013*), including the internal tissues of leaves in living plants (*Vaz et al., 2017*).

DNA metabarcoding is nowadays an essential tool in the methodological toolbox of fungal ecology, which has taken a monumental step forward since the advent of high-throughput DNA sequencing (*Brown, Leopold & Busby, 2018*). To our knowledge, the first study of DNA metabarcoding of foliar fungal communities, which included fungal endophytes, was in the phyllosphere of *Quercus macrocarpa*, a native tree species occurring in temperate climate (*Jumpponen & Jones, 2009*). Afterwards, since other pioneering studies in the beginning of this decade (*Zimmerman & Vitousek, 2012*; *Cordier et al., 2012*), many metabarcoding studies of foliar fungal endophytes have been performed (*Harrison & Griffin, 2020*).

The rapidly developing theory of complex networks, based on graph theory, has been successfully applied to uncover the organizing principles governing the formation and evolution of several complex biological systems (*Andrade et al., 2011*;

*Góes-Neto et al., 2010*). Bipartite interaction networks, which comprise interaction networks with two trophic levels, a lower and a higher, has been widely used to model two-level networks in Ecology, such as pollination and predator-prey (*Dormann et al., 2009*). Nonetheless, studies in natural ecosystems using DNA metabarcoding associated with bipartite interaction networks as a model to analyze foliar fungal endophytes-plant associations are still scarce (*Barge et al., 2019*; *Cobian, Egan & Amend, 2019*).

To date, as far as we know, there is no study investigating mycobiomes of *Myrtus communis*, the only European genus of Myrtaceae, using a metabarcoding approach modeled by bipartite networks. Additionally, the Mediterranean Forests, Woodlands and Scrubs, where *M. communis* is a bioindicator species, is one of understudied biomes of the world for fungal endophytes. Assuming the premises that (i) foliar endophytism may be an efficient strategy for saprotrophic fungi both as dispersal vehicle and as resource source during times of scarcity (*Nelson et al., 2020*) and (ii) a significant proportion of fungal endophytes of trees are saprotrophs (*Parfitt et al., 2010*), we hypothesize that, in biomes seasonally subjected to hydric deficiency, such as the Mediterranean biome, a high relative incidence and abundance of saprotrophic fungi will be inhabiting tree hosts when compared to pathotrophic and symbiothrophic guilds. Thus, in the current work, we performed, for the first time, a study on the spatial variation of fungal community structure in the foliar endophytome of this endemic plant of the Mediterranean biome, and their probable ecological functions in distinct individuals of *M. communis*, using bipartite network analysis as a model.

## MATERIALS AND METHODS

### Study area
Fieldwork was conducted in Sierra de Tejeda, Almijara y Alhama Natural Park. It is situated in the south of Andalusia, nearby the Mediterranean Sea, between Malaga and Granada provinces, and consists of several dolomitic mountain ranges. This park encompasses four distinct bioclimatic zones (thermo, meso, supra, and oromediterranean), with a wide variation in both mean annual temperature and rainfall, as well as elevation (*Pérez Latorre et al., 2004*).

### Sampling strategy and surface sterilization of the leaves
Following a water stream downhill, we established a 100 m long transect along this stream (South–North direction). The initial point of the transect corresponded to the geographical coordinates 36°51′25″N 03°41′40″W, with an elevation of approximately 360 m, corresponding to a subhumid to semiarid thermomediterranean bioclimatic zone (*Pérez Latorre et al., 2004*). Besides *Myrtus communis*, the following plants were commonly encountered in the sampling site: *Quercus suber, Quercus faginea, Smilax aspera, Pistacia lentiscus, Chamaerops humilis, Pteridium aquilinum, Brachypodium retusum, Dactylis hispanica, Phlomis purpurea, Rubus ulmifolius* and *Cistus salvifolius* (*Pérez Latorre et al., 2004*).

Five visually healthy leaves, with homogenous green coloration without any kind of discoloration or necrotic lesions, were collected from 11 visually healthy trees (without any

observable signs or symptoms of diseases), which were approximately separated 1.0 m from each other in the transect. Afterwards, all the samples were maintained in individualized sterile plastic bags and refrigerated until the surface disinfection procedure. The leaves were rinsed under running tap water to remove dirt and debris, and, subsequently, disinfected by successive dipping in 70% ethanol (1 min), 2% sodium hypochlorite (3 min) and sterile distilled water (2 min). Leaf fragments (5 mm$^2$) were excised from each leave in six specific positions: one from the base near the petiole, two from the middle vein, one from the left margin, one from the right margin, and one from the apex (*Saikkonen et al., 1998*; *Gamboa, Laureano & Bayman, 2003*). All the leaf fragments from each *M. communis* individual plants were then pooled and placed into 2-mL tubes with silica-gel in order to dehydrate and preserve the samples, and, thus, mitigating changes in the fungal diversity (*Vaz et al., 2014a*).

## DNA extraction, amplification and massively parallel sequencing

Leaf samples were ground with liquid nitrogen and 300 mg were used for genomic DNA extraction using the E.Z.N.A.® Plant DNA Kit Omega according to the manufacturer's instructions (Omega, Norcross, GA, USA). The quality and quantity of DNA were evaluated using spectrophotometry (NanoDrop ND 1000, NanoDrop Technologies, Wilmington, USA). After the extraction, the nuclear ribosomal internal transcribed spacer (ITS2) was amplified using the primers fITS7 (*Ihrmark et al., 2012*) and ITS4 (*White et al., 1990*). PCR amplification was performed using Kapa Taq DNA Polymerase High Fidelity Roche, Cape Town, South Africa) under the following conditions: 1 initial denaturation cycle at 94 °C for 2 min, followed by 35 denaturation cycles at 94 °C for 1 min, and annealing at 60 °C for 1 min, and extension at 72 °C for 3 min, with a final extension cycle at 72 °C for 5 min. At least three independent amplification reactions were performed from the same DNA extract. PCR products were then pooled in equimolar proportions based on their molecular weight and DNA concentrations, and purified using AMPure® Beads. The DNA was quantified using a fluorescence assay using Qubit®2.0 Fluorometer (Thermo, Waltham, MA, USA) and Qubit® dsDNA BR Assay Kit (Thermo, Waltham, MA, USA).

Sequencing libraries were generated using TrueSeq® DNA PCR-Free Sample Preparation Kit (Illumina, San Diego, CA, USA) following the manufacturer's recommendations, and index codes were added. The library quality was assessed on the Qubit@ 2.0 Fluorometer (Thermo Scientific, Waltham, MA, USA) and Bioanalyzer 2100 system (Agilent, Santa Clara, CA, USA). The library was sequenced on a MiSeq platform (Illumina, San Diego, CA, USA), and 2 × 250 bp paired-end reads were generated. All of the raw generated sequences were deposited in NCBI SRA under accession number PRJNA602325.

## Bioinformatic and ecological analyses

The output files (FASTQ format) of the amplicon metagenomic sequencing of each one of the samples comprise our raw primary data. The bioinformatics pipeline (Supplemental Material 1) was elaborated and run on an Operational System

Ubuntu 16.04.5 LTS system. The following programs were used: VSEARCH v2.9.1 (*Rognes et al., 2016*); BLAST v2.2.31+ (*Camacho et al., 2009*). Scripts in shell (*McIlroy, 1987*) and Python v3.0 (*Martelli, 2006*) programing languages were written to make some automatic tasks, such as merging samples or generating the abundance table. The reference database used for fungal taxonomic identification was UNITE v. 7.2 (*Abarenkov et al., 2010*). The pipeline comprised the following steps, all of which using VSEARCH and BLASTn, as aforementioned: (i) quality and length filtering was done with VSEARCH removing sequences smaller than 300 bp and default settings for quality filtering; (ii) dereplication was done with VSEARCH; (iii) detection and removal of chimeric sequences using the UNITE database (uchime_reference_dataset_untrimmed. fasta) and de novo implementation by VSEARCH); (iv) clustering sequences with similarity above 97% with VSEARCH; (v) automatic taxonomic identification with BLASTn was done in Python based in these rules, (Supplemental Material 2); and (vi) generation of the abundance table was built using python script (Supplemental Material 1).

Rows and columns within the interaction matrix represented distinct adult plant individuals of *Myrtus communis* (lower-level nodes) and fungal taxa (higher-level nodes), respectively. Each cell in the matrix comprised the number of reads of each fungal taxon in each *M. communis* individual. Therefore, the network was bipartite or two-mode in which nodes were divided in two disjoint sets and each link connected a node from one set (tree individuals) with a node from the other set (putative fungal species) (*Fodor, 2020*). In addition, the bipartite network was undirected, weighted, and quantitative. The R-package bipartite was used to visualize and plot the bipartite network and to calculate several network-and node-levels indices commonly used to describe patterns in bipartite ecological networks (*Dormann, Gruber & Fründ, 2008*) (Supplemental Material 3), and, specifically, MODULAR (*Marquitti et al., 2014*) was used to calculate the modularity (Q) by an annealing procedure to maximize Barber's modularity index (*Barber, 2007*). Moreover, classical bipartite network representation was generated using Gephi 0.9.2 (*Bastian & Heymann, 2009*).

The following aspects of network structure in both communitary and taxa levels were evaluated: connectance, nestedness, modularity, specialization, checkerboard score and generality. As the bipartite network of foliar fungal endophytes and *Myrtus communis* individuals was quantitative, when possible, the indexes were calculated in their weighted counterpart, such as weighted connectance, weighted NODF, H2' specialization, proportional generality and species strength (*Dormann et al., 2009*). The communitary (network) indexes were analyzed to investigate if there are common structural patterns in the foliar fungal endophytes - *Myrtus communis* individuals that could be also encountered in other bipartite networks of fungal endophytes - plants (*Mariani et al., 2019*), while the taxon (node) level indices were analyzed to discover the most important, relevant or influential fungal endophyte taxa in the studied interaction (*Marini et al., 2019*).

Fungal taxa were functionally classified into three ecological trophic modes (saprotrophic, pathotrophic and symbiotrophic, or a combination of these), and the fungal guilds of each of these trophic modes, using the FunGuild database (*Nguyen et al., 2016*).

**Table 1 Summary of the number of reads in the metagenomic pipeline.**

| Steps of analysis | Number of reads |
|---|---|
| Merge paired-end sequence | 212,167 |
| Shorten and/or filter the sequences | 33,364 |
| Merge strictly identical sequences | 20,538 |
| Pre-clustering the fasta sequences | 706 |
| Detect chimeras without external references (i.e., de novo) | 706 |
| Detect chimeras present with reference sequences | 599 |
| Extract all non-chimeric, non-singleton sequences, dereplicated (double-check) using Perl | 599 |
| Clustering | 599 |

As 11 fungal MOTUs (Molecular Operational Taxonomic Units) were not described in FunGuild, an extensive literature search was the strategy performed to describe the prevalent trophic mode of those MOTUs.

## RESULTS

The sequencing resulted in a total 621.9 Mb with 212167 reads for a total of 11 samples. Table 1 shows the number of reads after each step in our metagenomic workflow.

### Network structure and fungal endophyte diversity

The bipartite ecological network and corresponding adjacency matrix, comprising both the interactions between *Myrtus communis* individuals (lower trophic level) and the taxa of their foliar mycoendophytome (higher trophic level), are depicted in Figs. 1 and 2, respectively. The network had order ($N$) = 56 nodes and size ($M$) = 93 edges. As it is a bipartite network, the lower level (LL) was composed of 11 *Myrtus communis* distinct individual plants, and the higher level (HL) encompassed 44 different fungal taxa (MOTUs), as identified by bioinformatics analyses, followed by an extensive manual curation. Excluding those assigned as undefined or *incertae sedis*, fungal endophyte MOTUs were classified into 44 putative species in 28 genera, 23 families, 16 orders, 11 classes, five subphyla and two phyla (Table 2). The phylum Basidiomycota, and, specifically, the subclass Agaricomycotina, the class Agaricomycetes and the orders Polyporales, Hymenochaetales, and Agaricales were the most prevalent (relative MOTU richness) and the most frequent (relative read abundance) taxa in the higher trophic level (Table 2).

The bipartite network displayed an average of 1.7 links per node and an average linkage density of approximately 3.5 (Table 3). Weighted connectance was very low, nestedness (NODF, WNODF) was quite low, and modularity (Q) exhibited a medium value (Table 3). On the other hand, specialization (H2') and web asymmetry were moderately high (Table 3), mirrored in a high checkerboard score (C-score) of the higher trophic level (fungal endophytes).

The probability distribution of node degrees of the higher trophic level was very asymmetric. This asymmetry was reflected in a very high species strength of

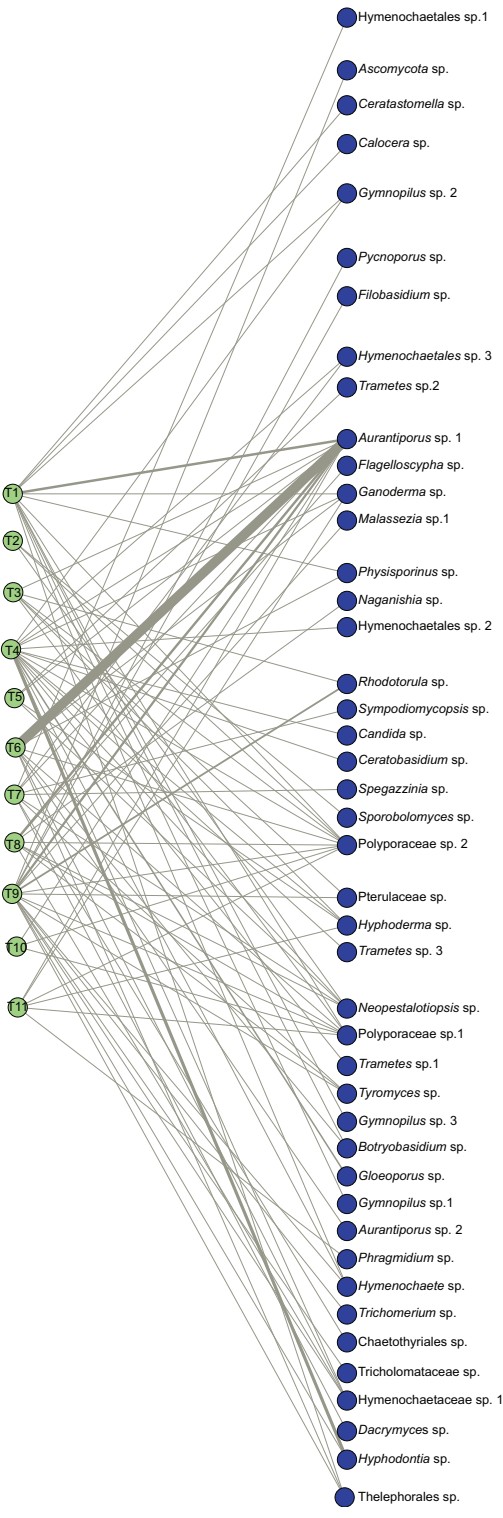

**Figure 1 Bipartite ecological network of *Myrtus communis* individuals and their foliar fungal endophytes.** The bipartite ecological network: green circles (nodes: tree set) represent *Myrtus communis* individuals (T1-T11) (left), and blue circles (nodes: fungal set) represent putative fungal endophytes species (right). Interacting taxa are linked by lines (links), whose width is proportional to the number of interactions.
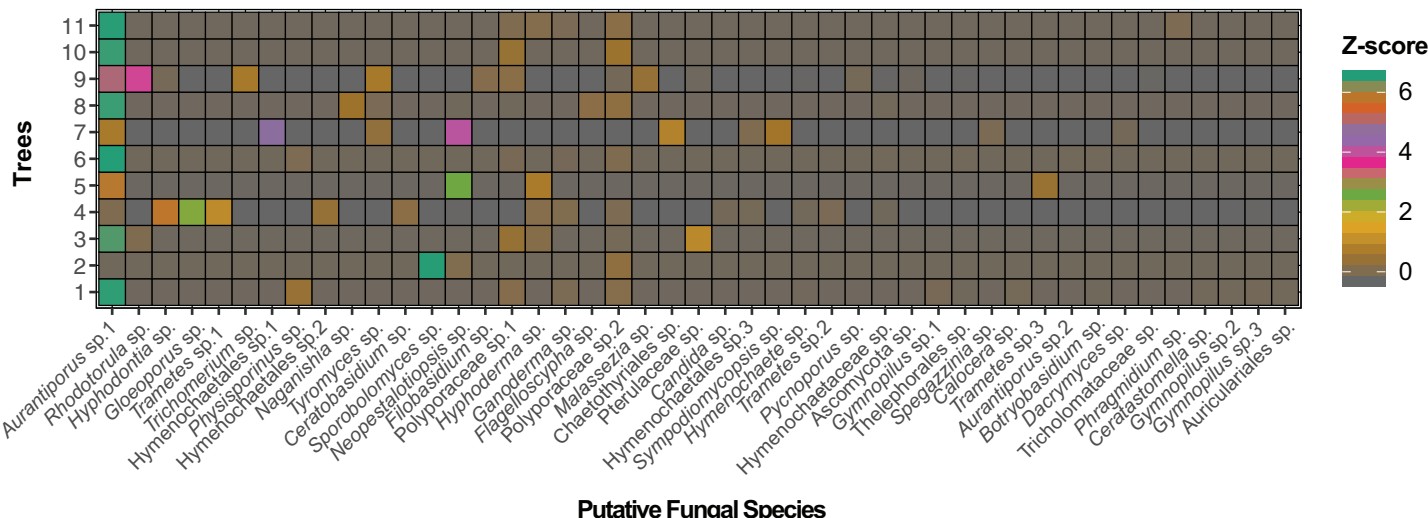

**Figure 2 Adjacency matrix of *Myrtus communis* individuals and their foliar fungal endophytes.** The adjacency matrix of bipartite ecological network with shading representing number of interactions per link, normalized by *z*-score.

*Aurantiporus* sp.1, which occurred in all but one *Myrtus communis* individuals. Furthermore, *Aurantiporus* sp.1 also displayed the highest weighted betweenness of all fungal taxa (Table 4), clearly showing the centrality and the importance of this fungal endophyte taxon in the network. Besides *Aurantiporus* sp., two other putative fungal species, Polyporaceae sp.1 and Polyporaceae sp.2, also exhibited a high proportional generality (Table 4), and the three MOTUs co-occurred in most of the sampled *Myrtus communis* individuals. Conversely, approximately 67% of putative species of foliar fungal endophytes displayed a very low value of proportional generality and, therefore, were qualitatively and quantitatively restricted to only one tree. Taken together, these results reinforced, even more, the asymmetric pattern of the higher-level taxa.

In lower trophic level (*Myrtus communis* individuals), the probability distribution of node degrees was much less asymmetric than in the higher trophic level (fungal endophytes), directly reflecting in a lower checkerboard score (C-score) and in a much higher niche overlap than those retrieved for the higher trophic level (Table 5). Apparently, there was no association between the distance of the sampled trees and their corresponding foliar endophytic fungi community since trees occurring more distantly were as similar as those that were nearer (e.g., trees no. 3, 10, and 11). Additionally, the trees that usually displayed the highest values of species strength and effective partners also exhibited the highest values of proportional generality (with few exceptions) (Table 5).

### Fungal endophyte trophic modes and guilds

The majority of putative fungal endophytes species (64.4%) were assigned to the class Agaricomycetes (Basidiomycota) (Table 2) and, except for the genus *Ceratobasidium*, all the other genera of detected Agaricomycetes, notably of the orders Polyporales, Agaricales, and Hymenochaetales were assigned as woody saprotrophs (Table 6). Moreover, a third of all the other putative endophytic fungal taxa of other classes,

**Table 2 Putative fungal endophyte species in *Myrtus communis* individuals.**

| Putative Species | Phylum | Subphylum | Class | Order | Family | Genus | % Identity | % Coverage |
|---|---|---|---|---|---|---|---|---|
| Ascomycota sp. | Ascomycota | Undefined | Undefined | Undefined | Undefined | Undefined | 95.25 | 98 |
| Aurantiporus sp. 1 | Basidiomycota | Agaricomycotina | Agaricomycetes | Polyporales | Meruliaceae | Aurantiporus | 99.38 | 100 |
| Aurantiporus sp. 2 | Basidiomycota | Agaricomycotina | Agaricomycetes | Polyporales | Meruliaceae | Aurantiporus | 98.70 | 94 |
| Botryobasidium sp. | Basidiomycota | Agaricomycotina | Agaricomycetes | Cantharellales | Botryobasidiaceae | Botryobasidium | 96.90 | 96 |
| Calocera sp. | Basidiomycota | Agaricomycotina | Dacrymycetes | Dacrymycetales | Dacrymycetaceae | Calocera | 97.15 | 100 |
| Candida sp. | Ascomycota | Saccharomycotina | Saccharomycetes | Saccharomycetales | Saccharomycetales | Candida | 98.71 | 97 |
| Ceratastomella sp. | Ascomycota | Pezizomycotina | Sordariomycetes | Incertae sedis | Barbatosphaeriaceae | Ceratastomella | 98.17 | 96 |
| Ceratobasidium sp. | Basidiomycota | Agaricomycotina | Agaricomycetes | Cantharellales | Ceratobasidiaceae | Ceratobasidium | 99.12 | 94 |
| Chaetothyriales sp. | Ascomycota | Pezizomycotina | Eurotiomycetes | Chaetothyriales | Chaetothyriales | undefined | 92.86 | 93 |
| Dacrymyces sp. | Basidiomycota | Agaricomycotina | Dacrymycetes | Dacrymycetales | Dacrymycetaceae | Dacrymyces | 99.66 | 95 |
| Filobasidium sp. | Basidiomycota | Agaricomycotina | Tremellomycetes | Filobasidiales | Filobasidiaceae | Filobasidium | 99.73 | 100 |
| Flagelloscypha sp. | Basidiomycota | Agaricomycotina | Agaricomycetes | Agaricales | Niaceae | Flagelloscypha | 99.19 | 95 |
| Ganoderma sp. | Basidiomycota | Agaricomycotina | Agaricomycetes | Polyporales | Polyporaceae | Ganoderma | 97.78 | 93 |
| Gloeoporus sp. | Basidiomycota | Agaricomycotina | Agaricomycetes | Polyporales | Meruliaceae | Gloeoporus | 99.29 | 91 |
| Gymnopilus sp. 1 | Basidiomycota | Agaricomycotina | Agaricomycetes | Agaricales | Strophariaceae | Gymnopilus | 97.66 | 100 |
| Gymnopilus sp. 2 | Basidiomycota | Agaricomycotina | Agaricomycetes | Agaricales | Strophariaceae | Gymnopilus | 98.24 | 94 |
| Gymnopilus sp. 3 | Basidiomycota | Agaricomycotina | Agaricomycetes | Agaricales | Strophariaceae | Gymnopilus | 98.25 | 100 |
| Hymenochaetaceae sp. | Basidiomycota | Agaricomycotina | Agaricomycetes | Hymenochaetales | Hymenochaetaceae | undefined | 95.38 | 100 |
| Hymenochaetales sp. 1 | Basidiomycota | Agaricomycotina | Agaricomycetes | Hymenochaetales | undefined | undefined | 95.67 | 98 |
| Hymenochaetales sp. 2 | Basidiomycota | Agaricomycotina | Agaricomycetes | Hymenochaetales | undefined | undefined | 90.49 | 96 |
| Hymenochaetales sp. 3 | Basidiomycota | Agaricomycotina | Agaricomycetes | Hymenochaetales | undefined | undefined | 91.10 | 97 |
| Hymenochaete sp. | Basidiomycota | Agaricomycotina | Agaricomycetes | Hymenochaetales | Hymenochaetaceae | Hymenochaete | 99.37 | 97 |
| Hyphoderma sp. | Basidiomycota | Agaricomycotina | Agaricomycetes | Polyporales | Meruliaceae | Hyphoderma | 96.95 | 100 |
| Hyphodontia sp. | Basidiomycota | Agaricomycotina | Agaricomycetes | Hymenochaetales | Schizoporaceae | Hyphodontia | 99.68 | 96 |
| Malassezia sp. 1 | Basidiomycota | Ustilaginomycotina | Malasseziomycetes | Malasseziales | Malasseziaceae | Malassezia | 99.02 | 100 |
| Naganishia sp. | Basidiomycota | Agaricomycotina | Tremellomycetes | Tremellales | Tremellaceae | Naganishia | 99.72 | 100 |
| Neopestalotiopsis sp. | Ascomycota | Pezizomycotina | Sordariomycetes | Xylariales | Amphisphaeriaceae | Neopestalotiopsis | 98.58 | 90 |
| Phragmidium sp. | Basidiomycota | Pucciniomycotina | Pucciniomycetes | Pucciniales | Phragmidiaceae | Phragmidium | 97.08 | 98 |
| Physisporinus sp. | Basidiomycota | Agaricomycotina | Agaricomycetes | Polyporales | Meripilaceae | Physisporinus | 98.51 | 100 |
| Polyporaceae sp. 1 | Basidiomycota | Agaricomycotina | Agaricomycetes | Polyporales | Polyporaceae | undefined | 95.04 | 94 |
| Polyporaceae sp. 2 | Basidiomycota | Agaricomycotina | Agaricomycetes | Polyporales | Polyporaceae | undefined | 95.29 | 94 |
| Pterulaceae sp. | Basidiomycota | Agaricomycotina | Agaricomycetes | Agaricales | Pterulaceae | undefined | 96.17 | 94 |
| Pycnoporus sp. | Basidiomycota | Agaricomycotina | Agaricomycetes | Polyporales | Polyporaceae | Pycnoporus | 99.09 | 100 |
| Rhodotorula sp. | Basidiomycota | Pucciniomycotina | Microbotryomycetes | Sporidiobolales | Sporidiobolaceae | Rhodotorula | 99.43 | 100 |
| Spegazzinia sp. | Ascomycota | Pezizomycotina | Dothidiomycetes | Pleosporales | Didymosphaeriaceae | Spegazzinia | 97.56 | 94 |
| Sporobolomyces sp. | Basidiomycota | Pucciniomycotina | Microbotryomycetes | Sporidiobolales | Sporidiobolaceae | Sporobolomyces | 99.12 | 100 |
| Sympodiomycopsis sp. | Basidiomycota | Ustilaginomycotina | Exobasidiomycetes | Microstromatales | Microstromataceae | Sympodiomycopsis | 99.71 | 97 |
| Thelephorales sp. | Basidiomycota | Agaricomycotina | Agaricomycetes | Thelephorales | unidentified | unidentified | 93.41 | 100 |
| Trametes sp. 1 | Basidiomycota | Agaricomycotina | Agaricomycetes | Polyporales | Polyporaceae | Trametes | 98.77 | 100 |
| Trametes sp. 2 | Basidiomycota | Agaricomycotina | Agaricomycetes | Polyporales | Polyporaceae | Trametes | 99.04 | 96 |
| Trametes sp. 3 | Basidiomycota | Agaricomycotina | Agaricomycetes | Polyporales | Polyporaceae | Trametes | 97.84 | 100 |
| Tricholomataceae sp. | Basidiomycota | Agaricomycotina | Agaricomycetes | Agaricales | Tricholomataceae | undefined | 85.68 | 94 |
| Trichomerium sp. | Ascomycota | Pezizomycotina | Eurotiomycetes | Chaetothyriales | Trichomeriaceae | Trichomerium | 97.89 | 100 |
| Tyromyces sp. | Basidiomycota | Agaricomycotina | Agaricomycetes | Polyporales | Polyporaceae | Tyromyces | 96.98 | 100 |

**Note:**
Complete taxonomic classification of putative fungal endophyte species in *Myrtus communis* individuals and their corresponding percentage coverage and identity.

**Table 3 Types and values of network-level indexes.**

| Index | Value |
|---|---|
| **Connectance** | **0.18989899** |
| **Web asymmetry** | **0.60714286** |
| **Links per taxa** | **1.67857143** |
| Nestedness | 21.5741105 |
| NODF | 29.5555935 |
| Weighted nestedness | 0.42947246 |
| **Weighted NODF** | **15.2729931** |
| Interaction strength asymmetry | 0.35324107 |
| **Linkage density** | **3.47731931** |
| **Weighted connectance** | **0.06209499** |
| **H2** | **0.61728682** |
| **Modularity (Q)** | **0.440754** |
| **Number.of.taxa.HL** | **44** |
| **Number.of.taxa.LL** | **11** |
| Mean.number.of.shared.partners.HL | 0.47777778 |
| Mean.number.of.shared.partners.LL | 2.65454545 |
| Weighted.cluster.coefficient.HL | 0.85253503 |
| Weighted.cluster.coefficient.LL | 0.33579411 |
| **Niche.overlap.HL** | **0.18578415** |
| **Niche.overlap.LL** | **0.47887023** |
| **C.score.HL** | **0.64785273** |
| C.score.LL | 0.39092274 |

Note:
Bold entries represent those indexes most explored in the characterization of the bipartite network.

regardless of pertaining to Basidiomycota or Ascomycota, generally showed a saprotrophic nutrition mode, which is mainly related to wood decomposition (Table 6). Altogether, the great majority of endophyte taxa in *Myrtus communis* foliar mycoendophytome comprised saprotrophs, and especially, woody saprotrophs (Fig. 3).

Approximately 36% of all fungal taxa were associated with pathotrophy either exclusively or in combination with saprotrophic (or, very rarely, symbiotrophic) nutrition mode due to less inclusive taxonomic assignment (Fig. 3). Pathotrophic fungal taxa were primarily related to the plant pathology guild; however, pathotrophy might be linked to animal pathogens, especially for the ascomycetous and basidiomycetous yeasts (e.g., *Candida* and *Malassezia*, respectively) (Table 6). In addition, exclusively symbiotrophic fungal endophytes were restricted to only one taxon (Fig. 3).

## DISCUSSION

This study modeled the fungal endophyte community diversity in leaves of *Myrtus communis* individuals as a bipartite ecological network. In this kind of network, every member of one trophic level is only connected to the members of the other trophic level, so that the interactions within trophic levels (lower or higher) are not represented for the

**Table 4 Types and values of higher trophic level indexes_.**

| Taxon | Degree | Normalized degree | Species strength | Weighted betweenness | Effective partners | Proportional generality |
|---|---|---|---|---|---|---|
| Ascomycota | 1 | 0.090909091 | 0.006877579 | 0 | 1 | 0.156384557 |
| Aurantiporus sp.1 | 10 | 0.909090909 | 6.000985121 | 0.69445883 | 4.513035865 | 0.705769113 |
| Aurantiporus sp.2 | 1 | 0.090909091 | 0.001449275 | 0 | 1 | 0.156384557 |
| Auriculariales | 1 | 0.090909091 | 0.005148005 | 0 | 1 | 0.156384557 |
| Botryobasidium | 2 | 0.181818182 | 0.004023278 | 0 | 1.889881575 | 0.295548292 |
| Calocera | 1 | 0.090909091 | 0.01029601 | 0 | 1 | 0.156384557 |
| Candida | 1 | 0.090909091 | 0.01346046 | 0 | 1 | 0.156384557 |
| Ceratastomella | 1 | 0.090909091 | 0.005148005 | 0 | 1 | 0.156384557 |
| Ceratobasidium | 1 | 0.090909091 | 0.037016265 | 0 | 1 | 0.156384557 |
| Chaetothyriales | 1 | 0.090909091 | 0.084175084 | 0 | 1 | 0.156384557 |
| Dacrymyces | 1 | 0.090909091 | 0.013468013 | 0 | 1 | 0.156384557 |
| Filobasidium | 1 | 0.090909091 | 0.027510316 | 0 | 1 | 0.156384557 |
| Flagelloscypha | 1 | 0.090909091 | 0.044414536 | 0 | 1 | 0.156384557 |
| Ganoderma | 4 | 0.363636364 | 0.067952092 | 0 | 3.044292689 | 0.476080362 |
| Gloeoporus | 1 | 0.090909091 | 0.198541784 | 0 | 1 | 0.156384557 |
| Gymnopilus sp.1 | 1 | 0.090909091 | 0.012870013 | 0 | 1 | 0.156384557 |
| Gymnopilus sp.2 | 1 | 0.090909091 | 0.005148005 | 0 | 1 | 0.156384557 |
| Gymnopilus sp.3 | 2 | 0.181818182 | 0.008843713 | 0 | 1.754765351 | 0.274418201 |
| Hymenochaetaceae sp.1 | 4 | 0.363636364 | 0.021985178 | 0 | 3.147762106 | 0.492261381 |
| Hymenochaetales sp.1 | 1 | 0.090909091 | 0.36026936 | 0 | 1 | 0.156384557 |
| Hymenochaetales sp.2 | 1 | 0.090909091 | 0.049915872 | 0 | 1 | 0.156384557 |
| Hymenochaetales sp.3 | 2 | 0.181818182 | 0.038151188 | 0 | 1.676575954 | 0.262190587 |
| Hymenochaete | 3 | 0.272727273 | 0.017014422 | 0 | 1.857027729 | 0.290410458 |
| Hyphoderma | 4 | 0.363636364 | 0.19456545 | 0 | 2.410325703 | 0.376937716 |
| Hyphodontia | 2 | 0.181818182 | 0.460574461 | 0.163127913 | 1.136984769 | 0.177806859 |
| Malassezia | 1 | 0.090909091 | 0.049518569 | 0 | 1 | 0.156384557 |
| Naganishia | 1 | 0.090909091 | 0.083445491 | 0 | 1 | 0.156384557 |
| Neopestalotiopsis | 5 | 0.454545455 | 0.612542425 | 0.100983946 | 2.005328881 | 0.313602468 |
| Phragmidium | 1 | 0.090909091 | 0.020100503 | 0 | 1 | 0.156384557 |
| Physisporinus | 2 | 0.181818182 | 0.090150243 | 0 | 1.999269584 | 0.312654887 |
| Polyporaceae sp.1 | 7 | 0.636363636 | 0.244426366 | 0 | 4.9200779 | 0.769424201 |
| Polyporaceae sp.2 | 9 | 0.818181818 | 0.342257352 | 0.041429311 | 6.264598218 | 0.979686415 |
| Pterulaceae | 2 | 0.181818182 | 0.139566226 | 0 | 1.865734575 | 0.291772074 |
| Pycnoporus | 1 | 0.090909091 | 0.014442916 | 0 | 1 | 0.156384557 |
| Rhodotorula | 2 | 0.181818182 | 0.322363942 | 0 | 1.064521894 | 0.166474784 |
| Spegazzinia | 1 | 0.090909091 | 0.02020202 | 0 | 1 | 0.156384557 |
| Sporobolomyces | 1 | 0.090909091 | 0.911764706 | 0 | 1 | 0.156384557 |
| Sympodiomycopsis | 1 | 0.090909091 | 0.067340067 | 0 | 1 | 0.156384557 |
| Thelephorales | 3 | 0.272727273 | 0.006123431 | 0 | 2.58640929 | 0.40447447 |
| Trametes sp.1 | 1 | 0.090909091 | 0.096466629 | 0 | 1 | 0.156384557 |
| Trametes sp.2 | 1 | 0.090909091 | 0.0625 | 0 | 1 | 0.156384557 |

| Table 4 (continued) | | | | | | |
|---|---|---|---|---|---|---|
| Taxon | Degree | Normalized degree | Species strength | Weighted betweenness | Effective partners | Proportional generality |
| Trametes sp.3 | 1 | 0.090909091 | 0.018508132 | 0 | 1 | 0.156384557 |
| Tricholomataceae | 1 | 0.090909091 | 0.002751032 | 0 | 1 | 0.156384557 |
| Trichomerium | 1 | 0.090909091 | 0.072902338 | 0 | 1 | 0.156384557 |
| Tyromyces | 3 | 0.272727273 | 0.132824122 | 0 | 1.858120562 | 0.29058136 |

Table 5 Types and values of lower trophic level indexes.

| Tree | Degree | Normalized degree | Species strength | Weighted betweenness | Effective partners | Proportional generality |
|---|---|---|---|---|---|---|
| T1 | 14 | 0.31111111 | 7.48918425 | 0 | 2.57524028 | 0.368943722 |
| T2 | 3 | 0.06666667 | 1.0355272 | 0 | 1.42560863 | 0.204240884 |
| T3 | 6 | 0.13333333 | 0.9258364 | 0 | 2.53708048 | 0.36347673 |
| T4 | 16 | 0.35555556 | 10.8434777 | 0.15686275 | 6.44529054 | 0.923389362 |
| T5 | 4 | 0.08888889 | 1.23405498 | 0 | 2.89831062 | 0.415228635 |
| T6 | 9 | 0.2 | 4.34350996 | 0.15686275 | 1.4668374 | 0.210147556 |
| T7 | 9 | 0.2 | 6.11079302 | 0.17647059 | 5.34577583 | 0.765866566 |
| T8 | 8 | 0.17777778 | 2.56345887 | 0.23529412 | 2.24831695 | 0.322106807 |
| T9 | 16 | 0.35555556 | 9.09088067 | 0.2745098 | 5.94181194 | 0.851258124 |
| T10 | 3 | 0.06666667 | 0.11981709 | 0 | 1.7172954 | 0.246029608 |
| T11 | 6 | 0.13333333 | 1.24345983 | 0 | 1.86340415 | 0.266961987 |

sake of simplicity (*Dormann, Gruber & Fründ, 2008*). Although using bipartite networks as a model to study animal-plant interactions is rather common (*Dormann et al., 2009*), studies specifically with endophytic fungi and their host plants are still quite rare (*Toju et al., 2014*).

The local bipartite network of *Myrtus communis* individuals and their foliar endophytic fungi is very low connected, with low nestedness, and moderately high specialization and modularity. Connectance is a commonly used indicator of complexity at network level, and it intuitively accounts for the probability that any pair of taxa interact in the network (*Landi et al., 2018*). In our *M. communis* individuals / endophytic fungi bipartite network, the connectance (which is a qualitative index) as well as its quantitative counterpart, weighted connectance, were low as well as other network level index measuring complexity, the nestedness and weighted nestedness. Connectance and nestedness are positively correlated (*Almeida-Neto et al., 2008*), and the low values retrieved in our bipartite network ecologically means that the great majority of fungal endophytes species occur in only one or very few *M. communis* individuals, and, conversely, very few fungal species occur in many plant individuals. This is clearly corroborated by not only the moderately high values of the network-level indexes of specialization (H2′) and checkerboard score (C-score), but also by the node-level indexes of node and normalized

Table 6 Ecological guilds of putative fungal endophyte species.

| Putative Species | Trophic Mode | Guild |
|---|---|---|
| Ascomycota sp. | Pathotroph–Saprotroph–Symbiotroph | All possible guilds |
| Aurantiporus sp. 1 | Saprotroph | Wood Saprotroph |
| Aurantiporus sp. 2 | Saprotroph | Wood Saprotroph |
| Botryobasidium sp. | Saprotroph | Wood Saprotroph |
| Calocera sp. | Saprotroph | undefined Saprotroph |
| Candida sp. | Saprotroph | Wood Saprotroph |
| Ceratastomella sp. | Pathotroph–Saprotroph–Symbiotroph | Animal Pathogen-Endosymbiont-undefined Saprotroph |
| Ceratobasidium sp. | Pathotroph | Plant Pathogen |
| Chaetothyriales sp. | Pathotroph–Saprotroph–Symbiotroph | Endomycorrhizal-Plant Pathogen-undefined Saprotroph |
| Dacrymyces sp. | Pathothroph | Endosymbiont-Plant Pathogen-undefined Saprotroph |
| Filobasidium sp. | Saprotroph | Wood Saprotroph |
| Flagelloscypha sp. | Saprotroph | undefined Saprotroph |
| Ganoderma sp. | Saprotroph | undefined Saprotroph |
| Gloeoporus sp. | Pathotroph–Saprotroph | Plant Pathogen-Wood Saprotroph |
| Gymnopilus sp. 1 | Saprotroph | Wood Saprotroph |
| Gymnopilus sp. 2 | Saprotroph | Wood Saprotroph |
| Gymnopilus sp. 3 | Saprotroph | Wood Saprotroph |
| Hymenochaetaceae sp. | Saprotroph | Wood Saprotroph |
| Hymenochaetales sp. 1 | Saprotroph–Symbiotroph | Ectomycorrhizal-Wood Saprotroph |
| Hymenochaetales sp. 2 | Pathotroph–Saprotroph–Symbiotroph | Ectomycorrhizal-Wood Saprotroph-Plant Pathogen |
| Hymenochaetales sp. 3 | Pathotroph–Saprotroph–Symbiotroph | Ectomycorrhizal-Wood Saprotroph-Plant Pathogen |
| Hymenochaete sp. | Pathotroph–Saprotroph–Symbiotroph | Ectomycorrhizal-Wood Saprotroph-Plant Pathogen |
| Hyphoderma sp. | Saprotroph | undefined Saprotroph |
| Hyphodontia sp. | Saprotroph | undefined Saprotroph |
| Malassezia sp. 1 | Saprotroph | undefined Saprotroph |
| Naganishia sp. | Pathotroph–Saprotroph | Animal Pathogen-undefined Saprotroph |
| Neopestalotiopsis sp. | Pathotroph–Saprotroph–Symbiotroph | Animal Pathogen-Endophyte-Epiphyte-undefined Saprotroph |
| Phragmidium sp. | Pathotroph | Plant Pathogen |
| Physisporinus sp. | Saprotroph | undefined Saprotroph |
| Polyporaceae sp. 1 | Saprotroph | Wood Saprotroph |
| Polyporaceae sp. 2 | Saprotroph | Wood Saprotroph |
| Pterulaceae sp. | Saprotroph | Wood Saprotroph |
| Pycnoporus sp. | Saprotroph | Wood Saprotroph |
| Rhodotorula sp. | Saprotroph | Wood Saprotroph |
| Spegazzinia sp. | Pathotroph–Saprotroph | Animal Endosymbiont-Animal Pathogen-Endophyte-Plant Pathogen-undefined Saprotroph |

| Table 6 (continued) | | |
| --- | --- | --- |
| Putative Species | Trophic Mode | Guild |
| Sporobolomyces sp. | Saprotroph | undefined Saprotroph |
| Sympodiomycopsis sp. | Pathotroph–Saprotroph | Fungal Parasite-Litter Saprotroph |
| Thelephorales sp. | Pathotroph | Plant Pathogen |
| Trametes sp. 1 | Symbiotroph–Saprothroph | Ectosymbiont-Wood Saprotroph |
| Trametes sp. 2 | Saprotroph | Wood Saprotroph |
| Trametes sp. 3 | Saprotroph | Wood Saprotroph |
| Tricholomataceae sp. | Saprotroph | Wood Saprotroph |
| Trichomerium sp. | Pathotroph–Symbiotroph | Ectomycorrhizal-Fungal Parasite |
| Tyromyces sp. | Symbiotroph | Endophyte |

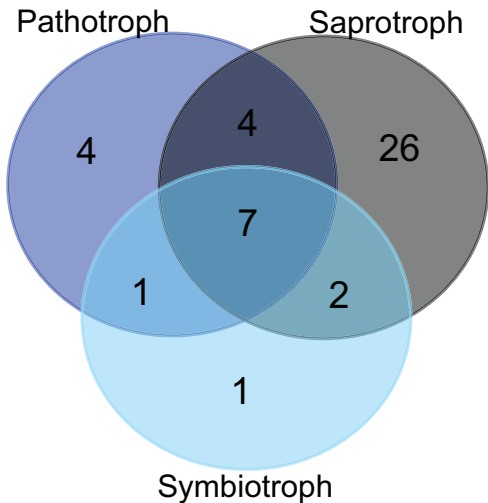

**Figure 3 Trophic modes of putative foliar fungal endophyte species.** Venn diagram representing the joint and disjoint occurrences of putative endophytic fungal species of saprothrophic, pathothrophic and symbiotrophic modes of nutrition in *Myrtus communis* trees.

degree, and effective partners and proportional generality. Furthermore, as there was no association between the distance of the sampled trees and their corresponding foliar fungal endophytes community and the environmental conditions are quite the same in sampling area, tree distance most probably serves or as proxy for dispersal limitation or this pattern occurs due to priority effects, and, therefore, are stochastic environmental drivers (*Amend et al., 2019*)

In spite of having worked with culture-dependent foliar endophytic fungal communities along with distinct North-American biomes, *Chagnon et al. (2016)* approximately retrieved these same network-level patterns. Furthermore, *Yao et al. (2019)* working on metabarcoding-analyzed endophytic fungal communities associated with the leaves of six mangrove species, observed that endophytic network structure was characterized by

significantly highly specialized and modular but lowly connected and anti-nested properties. Altogether, these similar findings pointed to possible common network-level patterns of foliar fungal endophyte communities in arboreous plants around the world.

The mycoendophytome of *Myrtus communis* individuals was, both qualitatively and quantitatively, mainly composed of basidiomycete fungi. Although there is a strong dominance of species of the phylum Ascomycota in culture-dependent methods (*Rashmi, Kushveer & Sarma, 2019*), this scenario drastically changed after the publishing of data originated from culture-independent and, more specifically, metabarcoding studies (*Yao et al., 2019*). These differences were mainly observed in the abundance of the commonly isolated genera and may be related to the ability of certain genera to grow readily on artificial media and overgrow other fungi (*Skaltsas et al., 2019*). Isolation of Basidiomycota in typical culture-based approaches, for instance, is challenging as they usually do not grow or develop slowly and are rapidly outcompeted by ascomycete species (*Martin et al., 2015*).

Although it is native of the Mediterranean biome, *M. communis* is a species cultivated worldwide for its medicinal uses (*Sumbul et al., 2011*). Therefore, different research articles have studied its relationships with fungi, many of which are found in the USDA fungal-host database, which keeps records for *M. communis* as early as 1941 (*Unamuno, 1941*), especially directly observable phytopathogenic fungi that were reported in that plant species. After revising the synonymy, using both Mycobank (http://www.mycobank.org/) and IndexFungorum (http://www.indexfungorum.org/) databases, there are 15 fungal species associated with *Myrtus communis* in their original region of occurrence, the Mediterranean biome, in the USDA Fungus-Host database. Furthermore, some of them are indeed basidiomycete woody saprotrophs, such as *Antrodia albida* and *Stereum reflexulum*. In our study, there are some taxa such as Polyporaceae sp.1/Polyporaceae sp.2 and Thelephorales sp. that could represent, for instance, a possible *Antrodia* and *Stereum* (such as cited in USDA fungus-host database), respectively, but we did not have a more inclusive and reliable taxonomic identification beyond family level. More recent studies, such as that of *Nicoletti et al. (2014)* reinforce the relationship between plant metabolism and fungal species content. Nonetheless, the common thread among all of those studies is the methodology, which is culture-dependent. It is well-known that those methods are context-dependent and highly influenced by the culture medium composition (*Stone, Polishook & White, 2004*), as well as by the fact that most of the microorganisms, including the fungi, are still unculturable using contemporary methods (*Hongoh, 2011*; *Salvioli et al., 2008*). Therefore, culture-dependent methods might be appropriate for single species-driven research, but highly underestimate taxonomic composition, richness, and abundance as in modern diversity studies (*Stefani et al., 2015*), which are conceivable using culture-independent high-throughput sequencing methods, such as the ones employed in our study. Although amplicon metagenomics is a large-scale, rapid, and independent of culturing and/or direct observation, as any method, there may also be biases, such as in the initial amplicon library preparation, differential primer annealing, PCR and sequencing artifacts, and contig assembly (*Brooks et al., 2015*; *Nilsson et al., 2015*).

Both methods; however, are complementary, and when possible, is very advantageous to use them in an integrative manner, especially to solve problems of species-level identification.

The Polyporales were, by far, the most prevalent and frequent foliar fungal endophytes of *Myrtus communis*, and, along with Hymenochaetales and Agaricales, accounted for 55.5% of all the putative endophytic fungi species. Even using a culture-based approach, in a study on a huge collection of native *Hevea brasiliensis* fungal endophytes, *Martin et al. (2015)* pointed out that 75% of all basidiomycete endophytes of this hyperdiverse tree of Amazon Forest encompassed species of the order Polyporales. In addition, Hymenochaetales and Agaricales also corresponded to a significant proportion of basidiomycete endophytes of the rubber tree (*Martin et al., 2015*).

Although it is not possible to state if there are any potential ecological benefit of these dominant wood-decomposing basidiomycete endophytes for *M. communis*, there are many reports in specialized literature accounting for positive impacts of fungal endophytes on their hosts, such as resistance to pathogens, herbivores, and abiotic stresses (*Delaye, García-Guzmán & Heil, 2013*). Nonetheless, the result of these interactions is highly context-dependent (*Saikkonen, Saari & Helander, 2010*). Amongst the Polyporales, the taxon *Aurantiporus* sp.1 dominated the foliar mycoendophytome of *Myrtus communis*. The genus *Aurantiporus* is rarely reported as an endophyte (*Dastogeer et al., 2017*) and comprises woody decomposers that produce white rot in their hosts and is frequently encountered in dead wood of angiosperms (*He et al., 2019*; *Zmitrovich, 2018*). *Aurantiporus* sp.1 is very closely related to *Aurantiporus* sp. KT156705, whose complete ITS sequence was derived from field-collected basidiomata on dead wood in Costa Rica, which, in turn, is more phylogenetically related to *Aurantiporus pulcherrimus* than any other species of the genus *Aurantiporus* (*Papp & Dima, 2018*). Nonetheless, multigene phylogenetic analyses showed that this genus has a polyphyletic origin, and, thus, a more detailed study is certainly required (*Papp & Dima, 2018*). Therefore, it is even possible that *Aurantiporus* sp.1 be a new species in this poorly studied genus.

As well as *Aurantiporus* sp.1, the great majority of the putative species of foliar fungal endophytes of *Myrtus communis* are saprotrophs, especially wood decayers, and a significant proportion was also categorized as pathotrophs. Actually, there is compelling evidence that endophytes could act as latent saprotrophs or latent pathogens (*Fesel & Zuccaro, 2016*; *Porras-Alfaro & Bayman, 2011*; *Schulz & Boyle, 2005*; *Sieber, 2007*). Thus, many foliar fungal endophytes would invade plant hosts either by leaf or even shoot surfaces and using, in the latter case, a sapwood route of infection (*Martin et al., 2015*; *Parfitt et al., 2010*).

Although the Foraging Ascomycete (FA) hypothesis was originally proposed more than 20 years ago (*Carroll, 1999*), it has only been tested quite recently, using the genus *Xylaria* in a tropical cloud forest site as a case study (*Thomas et al., 2016*). This hypothesis states that, for wood-degrading fungi, endophytism is a life-history strategy to span the scarcity of dead wood substrates and stressful environmental conditions, such as hydric restriction, in both time and space (*Carroll, 1999*; *Thomas et al., 2016*). Despite this FA hypothesis having been initially suggested for Ascomycota (*Carroll, 1999*), in fact, it can be

applied to any endophytic fungus regardless of its taxonomic assignment, and the term *viaphytism* has been recently proposed (*Nelson et al., 2020*). The Viaphytism hypothesis states that many fungi may be in a continuous and cyclical flux between life stages as endophytes in the forest canopy and as wood-decomposing fungi on the forest floor (*Thomas, Vandegrift & Roy, 2020*). Therefore, this cycle can really be a very common and still previously ignored ecological process in forests, which may have far-reaching implications for whole forest health (*Thomas, Vandegrift & Roy, 2020*). Thus, the dominance of basidiomycete woody saprothrophs in the foliar mycoendophytome of *Myrtus communis* in Southwestern Mediterranean sclerophyllous forest may be a possible adaptation of these wood-decaying fungi to cope with moisture limitation and spatial scarcity of their primary substrate (dead wood), which are totally consistent with the predictions of viaphytism hypothesis (*Nelson et al., 2020*; *Thomas, Vandegrift & Roy, 2020*).

## CONCLUSIONS

In conclusion, we carried out, for the first time, an amplicon (nrITS) metagenomic study on the spatial variation of the foliar mycoendophytome of *Myrtus communis*, an endemic tree of the Mediterranean biome, using, as a model, bipartite network analysis. The bipartite network of the trees and their foliar fungal endophytes was very low connected, and displayed low nestedness, and moderately high specialization and modularity. Similar communitary (network) patterns were also retrieved in both culture-dependent and metabarcoding of foliar endophytic fungi in distinct arboreal hosts in diverse biomes. Moreover, most of the putative endophytic fungi species were basidiomycete woody saprotrophs of the orders Polyporales, Agaricales and Hymenochaetales. Taking together, our findings corroborate the viaphytism hypothesis (*Nelson et al., 2020*; *Thomas, Vandegrift & Roy, 2020*), which states that saprotrophic fungi (especially the wood decayers) can utilize leaves both as dispersal vehicles and as resource during times of scarcity (*Nelson et al., 2020*). Furthermore, as the viaphytism hypothesis have been tested only in trees in humid biomes (rainforests) without any marked seasonal hydric deficiency (*Thomas et al., 2016*, *2019*), our study not only corroborate the viaphytism hypothesis but also extended it to a typical and endemic tree in Mediterranean biome, which is characterized by a dry and hot summer season that is very unfavorable to wood-decomposing fungi.

## ACKNOWLEDGEMENTS

The authors wish to thank Rafael León Morcillo to take the samples and Daniel Santana de Carvalho and Eric Roberto Guimarães Rocha Aguiar for helping with statistical analysis and image editing.

### Funding

Financial support for this study was provided by the Comissión Interministerial de Ciencia y Tecnología (CICYT), project (AGL2008-00572). Aline Bruna Martins Vaz has

received a scholarship from Coordenação de Aperfeiçoamento de Pessoal de Nível Superior (2010—Processo no 2330-10-5/CAPES) and CNPq (Conselho Nacional de Desenvolvimento Científico e Tecnológico of Brazil). José Siles and Inmaculada Sampedro have received support from the JAE program, which was co-financed by the Consejo Superior de Investigaciones Científicas (CSIC) and the European Social Funds, for the concession of a predoctoral grant (JAE-Predoc) to José Siles and a postdoctoral research contract (JAE-Doc) to Inmaculada Sampedro. The funders had no role in study design, data collection and analysis, decision to publish, or preparation of the manuscript.

### Grant Disclosures

The following grant information was disclosed by the authors:
Comissión Interministerial de Ciencia y Tecnología (CICYT): AGL2008-00572.
Coordenação de Aperfeiçoamento de Pessoal de Nível Superior: 2330-10-5/CAPES.
CNPq (Conselho Nacional de Desenvolvimento Científico e Tecnológico of Brazil).
Consejo Superior de Investigaciones Científicas (CSIC) and the European Social Funds.

### Competing Interests

Vasco Azevedo and Aristóteles Góes-Neto are Academic Editors for PeerJ.

### Author Contributions

- Aline Bruna M. Vaz conceived and designed the experiments, performed the experiments, analyzed the data, prepared figures and/or tables, authored or reviewed drafts of the paper, and approved the final draft.
- Paula Luize C. Fonseca performed the experiments, analyzed the data, prepared figures and/or tables, authored or reviewed drafts of the paper, and approved the final draft.
- Felipe F. Silva performed the experiments, analyzed the data, prepared figures and/or tables, authored or reviewed drafts of the paper, and approved the final draft.
- Gabriel Quintanilha-Peixoto performed the experiments, analyzed the data, prepared figures and/or tables, authored or reviewed drafts of the paper, and approved the final draft.
- Inmaculada Sampedro conceived and designed the experiments, performed the experiments, authored or reviewed drafts of the paper, and approved the final draft.
- Jose A. Siles conceived and designed the experiments, performed the experiments, authored or reviewed drafts of the paper, and approved the final draft.
- Anderson Carmo performed the experiments, authored or reviewed drafts of the paper, and approved the final draft.
- Rodrigo B. Kato performed the experiments, analyzed the data, prepared figures and/or tables, authored or reviewed drafts of the paper, and approved the final draft.
- Vasco Azevedo analyzed the data, authored or reviewed drafts of the paper, and approved the final draft.
- Fernanda Badotti performed the experiments, analyzed the data, authored or reviewed drafts of the paper, and approved the final draft.

- Juan A. Ocampo conceived and designed the experiments, authored or reviewed drafts of the paper, and approved the final draft.
- Carlos A. Rosa conceived and designed the experiments, prepared figures and/or tables, authored or reviewed drafts of the paper, and approved the final draft.
- Aristóteles Góes-Neto conceived and designed the experiments, performed the experiments, analyzed the data, prepared figures and/or tables, authored or reviewed drafts of the paper, and approved the final draft.

## Field Study Permissions

The following information was supplied relating to field study approvals (i.e., approving body and any reference numbers):

Sierras de Tejeda, Almijara y Alhama Natural Park.

## Data Availability

The data is available at NCBI SRA: PRJNA602325. Detailed bioinformatics analyses and bipartite network analyses are available in the Supplemental Files.

## Supplemental Information

Supplemental information for this article can be found online at http://dx.doi.org/10.7717/peerj.10487#supplemental-information.

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
