# Peer review of "Foliar mycoendophytome of an endemic plant of the Mediterranean biome (Myrtus communis) reveals the dominance of basidiomycete woody saprotrophs"

_PeerJ, doi:10.7717/peerj.10487_

## Round 0.1 · original submission · Major Revisions

The manuscript has been reviewed by two experts in the field, who raised important concerns that are very well described in their thorough reports. Although the reviewers recommended rejection, after carefully reading the manuscript and their reports, I feel that the authors deserve an opportunity to address the criticisms in a revised manuscript, as none of them irreversibly undermined the work. Therefore, my decision is for "major revision".

Reviewer 1 ·

Basic reporting

The authors use clear and professional English throughout the article. However, there are areas throughout the manuscript that can be cleaned up a bit. For example, in the Abstract on line 28 and 29, “The true myrtle, Myrtus communisis, is an evergreen perennial small tree that occurs in Europe, Africa, and Asia with a typical circum-Mediterranean geographic distribution.” This sentence could be cleaned up by describing the tree as a “small perennial evergreen tree” and removing the word “typical”.

I do not think this article provides sufficient background information to support a reason for conducting this research. The author's premise is that no one has ever used a “metabarcoding” approach to characterize the foliar fungal endophyte communities associated with this particular host nor has any one used network analysis to characterize the fungal community for this host. The majority of the introduction discusses the plant host. However, there needs to be more justification as to why this research is relevant and how it fits into the broader field of knowledge.

Their paragraph on endophytes (lines 72-78) needs to be expanded. There is a large body of literature on foliar fungal endophytes, but the authors only cited one article (Arnold 2008) in this paragraph besides their own articles (Vaz et al.). Their following paragraph (lines 79-87) which provides their reasoning for using high throughput amplicon sequencing over culture dependent techniques needs more background reference. I believe Zimmerman and Vitousek 2012 (PNAS, Fungal endophyte communities reflect environmental structuring across a Hawaiian landscape) was one of the first foliar fungal endophyte studies to use high throughput amplicon sequencing over culture dependent techniques.

Additionally, their paragraph on network analysis needs a lot more information and a more clear explanation of network theory and bipartite networks. In addition to Toju et al. (2014) article on fungal-plant interaction using bipartite networks, Cobian et al. (2019; ISMEJ, Plant-microbe specificity varies as a function of elevation) uses bipartite networks to study the effects of elevation on foliar fungal endophyte host specificity.

Figure 1: It would be more professional if the authors produced their own map using R or some GIS application instead of a Google Earth screenshot, additionally, it would be helpful for the readers if the transect line and sampling locations were shown on the map as well.

Figure 2: I would not recommend using this type of figure because they are very messy and difficult to interpret. A heat map as in Figure 3 may be a better way to show the fungal-plant interactions.

Figure 3: The quality of this figure needs to be improved. Also, it appears that there may be the same color green that represents 0 reads and somewhere around 2500 reads, this is the color of most of the squares. I would also suggest using a different color scheme since people with red-green color blindness would likely see no differences in these colors, and in fact, there really does not seem to be any differences in most.

The authors made the raw data available and accessible.

I do not think that the authors presented a clear hypothesis or question in their article. Therefore, the results of this article do not support any clearly stated hypothesis.

Experimental design

The research is within the Aims and Scope of PeerJ. However, there was no clear or well-defined research question or hypothesis in the introduction. Additionally, the methods were not clearly explained.

The authors do not indicate how many host plants they sampled or how far apart individual trees were from one another. They indicated that they collected five leaves per individual tree and then surface sterilized the leaves. However, they only used six leaf fragments after sterilization to extract DNA, and they do not explain the size of the leaf fragments. Also, they explain that the six leaf fragments were placed in silica-gel, but they do not provide a reason why. Finally, they do not indicate if or how leaf tissue was homogenized before DNA extraction.

The bioinformatics subsection in the Methods section needs a lot more detail. For example, lines 145-148 needs to explain how they did their quality filtering, how they performed chimera sequence removal, what percentage of sequence similarity did they use for clustering, etc. Also, they never reported the results of their bioinformatics which usually indicates the number of sequences they started with, how many sequences remained after quality control, how many OTUs were obtained after clustering.

For the bipartite networks (lines 150-158), the authors need to explain how their networks were constructed. Additionally, they need to explain the different indices they are testing and what the different indices tell us about the interactions between the fungi and host plants. This paragraph needs to be explained in much more detail.

As written, I do not thing the Methods section provided sufficient detail for someone to replicate this experiment.

Validity of the findings

Given the issues with this article, I am not providing any comments regarding the findings as I am uncertain of their validity.

Additional comments

When talking about foliar fungal endophytes, the authors use about three different terms. I would be a lot more helpful for the reader if the authors only chose one term.

Although the term “basidiomycotan” is not incorrect, I recommend using the term basidiomycete when referring to fungi within the basidiomycota.

·

Basic reporting

Well written and well referenced for the most part although problems arose in guild assignment of sequences, as discussed more fully under ‘Validity of Findings’.

Experimental design

The authors state that the foliar, endophytic fungi of Myrtus communis have not been studied with metabarcoding and bipartite network analysis. So, they undertook to fill this gap. They were not testing a particular hypothesis that could have been falsified by their findings. Methods are described clearly enough, although local plant communities should have been noted, for reasons discussed below.

Validity of the findings

General Comments: I first compared their list of fungal taxa, mostly genera, with the list of 33 non-synonymous genera obtained for Myrtus communis from a search of the USDA SMML fungal databases. Myrtus communis is grown widely in many parts of the world so the 33 genera represent global records. None of the 33 were found as foliar endophytes in this study! This complete lack of overlap was surprising although it passed without comment from the authors. I then started to examine the authors’ guild assignments of Table 5 more closely. The first thing to strike me was their finding of Phragmidium which the authors assigned, via FUNGuild apparently, to the ‘saprotroph’ category. That is just wrong; Phragmidium is a genus of rust fungi, obligate, biotrophic parasites, that attack plants in the Rosaceae. Phragmidium is not known to successfully parasitize Myrtus communis. An explanation of its amplification in this study must thus involve a source of its spores (infected Rosa species?) that might have been near the sampled Myrtus communis. As we know from the work by Michelle C. Heath and others, spores of rust fungi can germinate on a non-host and even substomatal space can be colonized. That level of colonization would not be washed off in reported protocols. If that is the explanation for the Phragmidium finding here, then one would have to question additional findings since other plants could also have been sources of the other reported fungi. I also wondered why the finding of Nicoletti et al was not duplicated here, and passed without comment from the authors: Neofusicoccum australe as endophyte in M. communis.
Regarding the authors’ conclusions, I was unconvinced not only due to my doubts about their findings. A second concern was that their ‘hypothesis’ could not have been rejected if their conclusion were “in accordance with the hypothesis stating that the endophytism is a life-history strategy” of whichever fungi were found.

Additional comments

"Culture-dependent" work is mentioned in the Abstract but no Results are reported.

---

## Round 0.2 · Major Revisions

The manuscript has addressed most of the points raised by the reviewers. Nevertheless, one of them still have some criticisms to be addressed.

·

Basic reporting

No comment

Experimental design

No comment

Validity of the findings

In their rebuttal, the authors "agreed with the Reviewer 2 and discussed this complete lack of overlap based on the very strong selection caused by traditional isolation methods based on standardized culture media for fungi. Please, see Lines: 314-328 in the Discussion." I don't find this argument convincing because many of the fungi that they are reporting here (e.g., Aurantiporus) are as culturable as the fungi and oomycetes reported in the USDA SMML databases for Myrtus communis. It is also misleading to call the SMML fungi and oomycetes 'culture-dependent'. They are culturable but they do not depend on culture to affect Myrtus communis in ecologically significant ways. The authors report Phragmidium, a rust fungus that has no ecologically significant role in M. communis (and the DNA of which may have come from neighbouring Rubus), but do not report Austropuccinia psidii, a rust fungus that does attack M. communis, at least in some parts of the world. Was there heart rot caused by Aurantiporus in the trees sampled for leaf endophytes? Or, could spores from basidiomata of Aurantiporus on nearby Acer or Populus or Salix (etc.) have blown over and stuck to the Myrtus leaves as the Phragmidium spores evidently did? If Phragmidium was just trace DNA and not ecologically meaningful in leaves of M. communis, then could not this also be true of other reported taxa in Table 6?

---

## Round 0.3 · Major Revisions

Both reviewers raised some concerns that should be addressed in a revised manuscript, in particular regarding the clarity in some sections and on how the problem addressed in the manuscript is presented.

Reviewer 3 ·

Basic reporting

Pag. 2, Line 67: The authors should consider describing better the what is a myrtucommulones, because it is not clear in the sentence.
Pag. 3 Line 109: Remove one of the words: with or since from the sentence.
Pag 5. Line 156: I believe that the correct term to be used would be disinfested.
Pag. 7 Line 230: Replace 212,167 for 212.167.
Pag. 7 Lines 241-244: The english language here should be improved. Using the words “not only” and “but also” made the sentence very confusing. Please re-write.
Pag. 9 Line 335: The citation Thomas et al., 2012 is not in the reference section. Also, you said our study and cites another author, please review that.
Pag. 9 Lines 331-335: Considering that your work highlight the benefits of using culture-independent high-throughput sequencing methods instead of culture-dependent methods, but in Pag. 10 Lines 343-353 is suggested that the genus Aurantiporus sp. found in this work could belong to two different species or an entirely new species. So, don’t you think that this represents a lack of a precise identification that could be benefited from a study using a culture-dependent method? Maybe we should not prefer one technique over another but use both techniques to obtain a more precise and detailed work.
Pag. 10 Lines 356-358: The english language here should be improved. Using the words “there has long been” made the sentence very confusing. Please re-write.

Experimental design

I believe the authors should make their justification clearer in the text, since I was little confused about the purpose of the work. The objective of the work was to describe a fungal community diversity in a plant of medicinal interest in an area that is endangered and little studied? Because fungal endophytes are important for plants? Or was it to demonstrate the efficacy of a metagenomic technique? How all of this is connected?

Validity of the findings

The authors propose two questions in the end of the introduction section, which are:
(i) Are the bipartite network-level patterns of Myrtus communis-foliar fungal endophytes similar to the networks of other foliar fungal endophytes and their distinct hosts in different biomes? and (ii) Are the fungal trophic modes and guilds related to the particular bioclimatic features of the Mediterranean sclerophyllous forests ecoregion?
However, the first question was answered very indirectly, and the second question was not answered in the discussion or in the conclusions section. Please review that.

Additional comments

This is an interesting work with important metagenomic data. The paper is well written, and without major errors in relation to the English grammar. The results were clearly described, the discussion is well based, and the literature is updated. However, the authors proposed to answer two questions, but they answered both questions in an indirect and inconclusive way. Also, the paper seems very descriptive, since the authors does not provide a clear justification for the importance of this type of work. For this reason, it would be interesting for the authors to explain the purpose of the research more clearly in the text and even provide suggestions on how they could expand this research by using other types of techniques and performing other analysis.

Reviewer 4 ·

Basic reporting

The topic addressed in this study is interesting, but as it is presented, it is not new. In the introduction and discussion, a lot of information is missing regarding the novelty of the work. The introduction is not well structured; as suggestion, first briefly describe the broad research and then narrow down to your particular topic. Sometimes the text is also not well linked; for example, the paragraph (lines 79-87) is absolutely not related with the previous sentence about fungal endophytes. My second major concern is regarding the main goal and the ecological novelty of the study. Clear objectives of the study are missing here. Furthermore, I don´t see any ecological novelty in this work. It´s not clear what ecological question the authors seek to answers by using this bipartite networks, and what´s novel of that. Bipartite networks are usually used to represent ecological communities. But the authors here just worked with one plant species. It would have been nice to see the structure of fungal endophyte communities along an array of plant species in this Mediterranean biome. The article is in general well written and clear to follow.

Experimental design

Methods are described with enough details. Authors are using novel bioinformatic data here; nevertheless I miss a nice ecological hypothesis in the introduction, which properly justify the use of bipartite networks in this study.

Validity of the findings

- Results are clear and well explained. But to the lack of an ecological hypothesis in the introduction, it´s difficult to see the novelty of the results.
- I miss a deep discussion of the results as well as the biological relevance of them. For example, what´s the ecological relevance in this kind of interactions that individuals of M. communis and foliar endophytic fungi are very low connected, with low nestedness, and moderately high specialization? Findings should be set the into the context of the literature, and then into broader theory.
- It would be nice that authors, together with weaknesses of culture-dependent methods, add some information about weaknesses of a metagenomic approach.
- The importance and novelty of this research with respect to the existing ones should be highlighted. Similarly, the potential importance/benefits of dominant fungal endophytes for the studied plant species should be included. This aspect, although speculative, should be included in the discussion.
- Figure 1 should be moved to supplementary material.

---

## Round 0.4 · accepted · Accept

The authors have addressed all the concerns raised by the reviewers. Therefore, I recommend the acceptance of this manuscript.

Reviewer 3 ·

Basic reporting

no comment

Experimental design

no comment

Validity of the findings

no comment

Reviewer 4 ·

Basic reporting

I thank the authors for the corrections made; the manuscript improved markedly.
I'm still not completely convinced with the structure of the introduction; nevertheless it improved regarding the last version. The discussion has also considerably improved, showing the ecological relevance of the data. The new version of the manuscript highlights much better the novelty of the work, and it is in general very well written.

Experimental design

Methods are described with enough details.

Validity of the findings

Results are clear and well explained.